# Evaluation of the Pragmatic Implementation of a Digital Health Intervention Promoting Healthy Nutrition, Physical Activity, and Gestational Weight Gain for Women Entering Pregnancy at a High Body Mass Index

**DOI:** 10.3390/nu15030588

**Published:** 2023-01-22

**Authors:** Shelley A. Wilkinson, Brianna Fjeldsoe, Jane C. Willcox

**Affiliations:** 1School of Human Movement and Nutrition Sciences, The University of Queensland, St Lucia, QLD 4072, Australia; 2Enable Health Consulting, Brisbane, QLD 4006, Australia; 3Centre for Quality and Patient Safety, Institute of Health Transformation, Deakin University, Burwood, VIC 3125, Australia; 4Impact Obesity, South Melbourne, VIC 3205, Australia

**Keywords:** pregnancy, weight, nutrition, implementation, digital, physical activity, facilitators, barriers, intervention

## Abstract

txt4two is a multi-modality intervention promoting healthy pregnancy nutrition, physical activity (PA), and gestational weight gain (GWG), which had been previously evaluated in a pilot randomized controlled trial (RCT). This study aimed to evaluate a pragmatic implementation of an adapted version of txt4two in a public tertiary hospital. Using a consecutive cohort design, txt4two was delivered to women with a pre-pregnancy BMI > 25 kg/m^2^, between 10 + 0 to 17 + 6 weeks. Control and intervention cohorts (n = 150) were planned, with surveys and weight measures at baseline and 36 weeks. The txt4two cohort received a dietetic goal-setting appointment and program (SMS, website, and videos). The navigation of disparate hospital systems and the COVID-19 pandemic saw adaptation and adoption take two years. The intervention cohort (n = 35; 43% full data) demonstrated significant differences (mean (SD)), compared to the control cohort (n = 97; 45% full data) in vegetable intake (+0.9 (1.2) versus +0.1 (0.7), *p* = 0.03), fiber-diet quality index (+0.6 (0.8) versus 0.1 (0.5), *p* = 0.012), and total diet quality index (+0.7 (1.1) versus +0.2 (±0.6), *p* = 0.008), but not for PA or GWG. Most (85.7%) intervention participants found txt4two extremely or moderately useful, and 92.9% would recommend it. Embedding the program in a non-RCT context raised implementation challenges. Understanding the facilitators and barriers to adaptation and adoption will strengthen the evidence for the refinement of implementation plans.

## 1. Introduction

Suboptimal antenatal health behaviors, including diet quality and physical activity, and excess gestational weight gain (GWG) are associated with negative pregnancy-related and long-term health outcomes for both the mother and offspring. These include the increased risk of caesarean sections [1,2], pre-term birth [1,2], and offspring’s risk of chronic disease in adult life [2,3,4,5]. Pregnant women who gain weight in accordance with the US Institute of Medicine (IOM) guidelines, also adopted for use in Australia [1,6,7,8], demonstrate a lower risk of pregnancy and birth-related complications, compared with women who gain above or below the GWG guidelines. 

In Queensland, Australia, between 30% to 100% of women (depending on their body mass index (BMI) category) gain more weight than recommended during pregnancy [9,10,11]. A total of 30% to 50% of women are overweight or obese at the beginning of pregnancy, placing them at increased risk of excess GWG, with its associated complications [9,10,11,12,13]. Very few women meet the pregnancy fruit and vegetable guidelines (~10% and ~5%, respectively) or undertake sufficient physical activity (40%) [9,14]. A high BMI and a poor lifestyle also increase a woman’s risk of developing gestational diabetes mellitus (GDM) [15]. 

A recent systematic review and meta-analysis noted that weighing, as a stand-alone intervention, does not reduce excess GWG [16]. However, it has been shown that interventions that include dietary advice and physical activity, supported by ongoing weight monitoring, can prevent excess GWG and result in more women, across all BMI categories, achieving weight gain within the recommended ranges [17]. This approach has been shown to be efficacious when delivered by obstetricians and midwives and supported by dietitians in an antenatal setting [18].

Since the opening of the new Mater Mothers’ Hospital (MMH) in Queensland, Australia, in 2008, a number of service delivery initiatives have been developed, implemented, and evaluated to meet the identified nutrition needs of women attending the service [9]. Following iterative improvements in numerous knowledge-to-action cycles, the progressively evolving MMH nutrition and dietetic service has included service mapping and targeted service improvements based on identified evidence–practice gaps [9,19,20,21,22,23,24]. Through ongoing service monitoring, the programs have resulted in improvements in staff knowledge, attitudes, and behaviors, regarding the management of GWG [22,25], and, for women attending the service, statistically and clinically significant increases in servings of fruit (+0.4 serves/day, *p* = 0.004) and vegetables (+0.4 serves/day, *p* = 0.006), compliance with fruit guidelines (+11.9%, *p* < 0.001), and the diet quality score (*p* = 0.027) [19] (the “Healthy Start to Pregnancy” (HSP) program), as well as significant shifts in the proportion of women gaining within the recommended GWG guidelines (by 15.4%) and a reduction of excessive gain (by 24.7%, *p* < 0.001) over time [26,27]. However, diet quality in many women in pregnancy remains poor, a proportion of women continue to gain weight outside recommended GWG ranges for their pre-pregnancy body mass index (ppBMI), and access to services remains inequitable [26].

The next service priority for the MMH was the development of personally tailored strategies to support women to improve diet quality and decrease excess GWG. This priority is to be realized through exploring the potential of broad reach/low intensity methods of delivery, such as digital health approaches. When these approaches are embedded in maternity service, they can offer the opportunity for tailored messages that provide information and resources to facilitate knowledge acquisition and behavior change. This approach may help to clarify the complexity women experience from the ‘noise’ of social media and traditional media by offering evidence-based messages from trusted health services in more accessible modalities [28,29]. Rather than developing a new digital health program, MMH opted to evaluate the implementation of an adapted, previously tested intervention. 

The program to be adopted and adapted is the txt4two program [30], chosen due to its cultural alignment with co-design and development in-country and the availability of the program content from the developer (JW). txt4two was evaluated in a pilot randomized controlled trial (RCT) in an Australian tertiary maternity hospital to determine its feasibility and effectiveness in promoting a healthy diet, physical activity, and GWG in pregnant women [30,31]. One hundred women who were overweight or obese prior to pregnancy were recruited at their first hospital antenatal visit and randomized to receive usual care or usual care plus the txt4two program. The primary outcome was intervention feasibility, and the secondary outcomes were objectively measured changes in GWG and self-reported dietary intake and physical activity. Formative work [32,33,34] was used to develop content, message phrasing, and timing, and txt4two had a strong theoretical underpinning of behavioral techniques [35]. Delivery to protocol provides evidence of program feasibility. Most women engaged regularly with the program, with the majority (97.6%) reporting that the intervention was helpful in improving individual health behaviors. Secondary outcomes demonstrated a significantly lower GWG in the intervention group (7.8 (±4.7) kg versus 9.7 (±3.9) kg; *p* = 0.041), compared with the control group at intervention completion. The intervention group women reported significantly smaller reductions in physical activity from baseline to completion of the intervention (*p* = 0.001), compared with the control group, but no differences in consumption frequencies of key food groups. The pilot RCT concluded that txt4two was feasible to implement and produced positive physical activity and GWG outcomes [30]. Eighty percent of the goals set at the introductory session were related to physical activity habits [30]. 

The aim of this current study was to evaluate the pragmatic implementation of an adapted and site-specific version of the txt4two program into the MMH antenatal nutrition and dietetic service. The outcomes provide further understanding of implementing an intervention in a system with a different service infrastructure to the original research context [36]. 

## 2. Materials and Methods

### 2.1. Study Design

This study followed a consecutive cohort design. It was a prospective, pragmatic implementation and evaluation informed by the RE-AIM framework [37] and the standards for quality improvement reporting excellence (SQUIRE) guidelines [38]. Approval was obtained from Mater Misericordiae Ltd. Human Research Ethics Committee (HREC/MML/47416). All methods were performed in accordance with the relevant guidelines and regulations (e.g., Declaration of Helsinki). The trial was registered with the Australia New Zealand Clinical Trials Registry (ACTRN12619000004167).

### 2.2. Participants and Recruitment

This study was delivered through the Brisbane MMH antenatal clinic across 2019–2020. Eligible women were women who entered pregnancy with a BMI ≥ 25 kg/m^2^, were receiving publicly-funded MMH antenatal care, had a singleton, live gestation between 10 + 0 and 17 + 6 weeks, and who owned a mobile phone. Women were excluded if they met one or more of the following criteria: receiving privately-funded MMH antenatal care, multiple pregnancy, comorbidities requiring significant medical and dietary management; discontinuation of care at hospital, or insufficient English to understand the intervention.

Eligible women were identified through numerous channels, i.e., the MMH online registration form completed by some women attending the publicly-funded antenatal services, recruitment advertisements in the MMH antenatal clinic and/or social media accounts (Instagram and Facebook), invitation by antenatal health professionals (dietitian, obstetrician, and midwife), a link provided on the Mater Research Institute’s clinical trials site, and direct SMS invitation to participate during the hospital registration process. 

An initial prospective control (PC) group who received usual care (advertised as the ‘Healthy Living in Pregnancy’ survey for interest, HLIP) was planned to be recruited (n = 150). Following this, it was planned that women (n = 150) would be recruited into the txt4two intervention. Following consent, details were then provided to administrative services to book women into the MMH dietitian’s antenatal clinic. Women who chose not to participate in the txt4two program received antenatal care, including dietetics referrals and reviews, as clinically appropriate. 

### 2.3. Sample Size

Based on detecting the ~2 kg between group difference (+4.3) in weight gain from the pilot RCT [30], we needed 119 women per group to provide 80% power. Assuming a 10–20% drop out between baseline and follow-up, a recruitment target of 150 women per group was planned. 

### 2.4. Intervention—The txt4two Program

The txt4two intervention was delivered from baseline (i.e., between 10 + 0 and 17 + 6 weeks) to 36-weeks’ gestation and consisted of a tailored suite of theoretically grounded, evidence-based intervention strategies focusing on promoting healthy nutrition, physical activity, and GWG. Intervention modalities included: bi-directional text messages (SMS) promoting positive health behaviors, goal setting and self-monitoring, video messages, and an information website.

The txt4two intervention content was developed according to evidence-based guidelines. The Institute of Medicine GWG guidelines [6] provided the GWG recommendations. The nutrition content was based on the recommendations of the Australian Dietary Guidelines for pregnancy [7], with emphasis on replacement of sugar sweetened beverages, increased fruit and vegetable intake, reduction of discretionary food groups, and consumption of regular meals. The physical activity components were based on national guidelines for pregnancy [39]. The emphasis was on 30 min of moderate intensity physical activity on most, if not all, days of the week, reduction of sedentary behaviour, and abdominal and pelvic floor strengthening. The behaviour change guidance was informed by the CALO-RE taxonomy of behaviour change techniques [40]. 

The process of adapting and embedding the program into the MMH service commenced with system-level work to identify key representatives and systems to involve in stakeholder engagement and ensure ownership and suitability for the local site. The adaptation of the txt4two model was systematically planned with a proactive process of modification to fit the intervention into the MMH context and enhance its feasibility and acceptability [36]. The delivery of txt4two mirrored that of the initial pilot, with four exceptions: (1) MMH dietitians, rather than a trained research assistant, introduced the program, (2) the social media community group was excluded, (3) women volunteered to enter the study, rather than being approached in person after screening, and (4) all weight outcomes were self-reported [30]. Three different modalities were offered to women to introduce the program and set goals, and women were asked to rank their first, second, and third preferences. The introductory session modalities offered were: face-to-face (F2F) visit with a MMH dietitian; group F2F workshop led by MMH dietitian (HSP,19); or telehealth delivery by a MMH dietitian. Regardless of the delivery format, at each introductory session (after baseline assessments), a trained dietitian explained the intervention and provided a booklet outlining the txt4two program, weight tracking, and goal setting. In addition, the dietitian discussed appropriate GWG targets and individual GWG monitoring and recording and asked the woman to set a nutrition or physical activity goal to work towards the evidence-based recommendations. 

Following the introductory session (which included goal setting around nutrition and physical activity habits that could be changed every six weeks), intervention participants received four to five individually tailored interactive SMS per week. The SMS delivered information specific to the individual’s gestational week, encouragement of positive health behaviors, monitoring of individual goals, and encouragement of self-monitoring of GWG. Original texts developed and mapped according to the behaviour change techniques by the authors J.W. and B.F., and they were adapted by S.W. through wide stakeholder consultation. Women chose the frequency of texts that aimed to: prompt review of their weight (weekly or fortnightly) and check their behavioral goals (weekly or fortnightly). SMS receipt was confirmed by telephone early in the intervention. As with the previous study, the SMS were bi-directional, allowing women to provide feedback on their goal progress and/or weight, triggering different pre-programmed responses of positive reinforcement or encouragement.

A study-specific website outlined intervention content information (https://www.matermothers.org.au/antenatal-and-pregnancy-services/nutrition-and-dietetics/txt4two-program (1 February 2019)). Short videos featuring an obstetrician, dietitian, or physiotherapist were embedded on the website, outlined the benefits of the intervention, and explained the components. The website was promoted and linked in the SMS content. The SMS were administered via a web-based platform, propelo™ (https://propelo.com.au).

### 2.5. Compensation

Women recruited to the control arm of the study were invited to enter the draw to win one of three Mater Mothers’ product gift packs (AUD 31.98 RRP) at the completion of the baseline and follow-up surveys. Women from the intervention arm were offered the chance to go into the draw to receive one of three AUD 50 department store gift cards on completion of each survey.

### 2.6. Outcome Measures and Data Analysis

To meet the project aims, a comprehensive evaluation, based on the RE-AIM model, was planned [37]. Specifically, we were interested in implementation/adoption, reach, and effectiveness elements, as appropriate for the intervention. 

Implementation/Adoption: Barriers and facilitators to embedding the program into existing services for delivery were interrogated, with examination of study records for timelines and MMH system interactions. Further, senior author (JW) led a discussion with the project team about the facilitators and barriers to the adoption of a digital program within the MMH systems. These discussion data were analyzed thematically. 

Reach: The proportion of women willing to take part in the txt4two program and their representativeness, compared to the broader MMH population. A comparison of the study population’s demographics and anthropometrics (age, education level, and ppBMI) was compared with the wider MMH public-hospital population during the study period. Participation rates were also calculated from MMH activity data during the study period. 

Effectiveness: In changing appropriate maternal GWG, fruit and vegetable intake and diet quality, and changing physical activity levels, as measured for women enrolled in the txt4two intervention, compared with: (a) the prospective control group who did not to receive txt4two (as the program was not implemented in the clinic when the women were recruited), but completed two surveys, and (b) between modalities of the introductory session of the txt4two program (i.e., F2F, Group F2F, or telehealth). 

All women were sent a URL via text message to a survey hosted by Survey Monkey at baseline and 36 weeks’ gestation. The survey requested socio-demographic, (age, education, income, aboriginal and/or Torres Strait Islander status), pregnancy, and anthropometric characteristics and dietary quality and physical activity via valid questionnaires. Women were asked their height, pre-pregnancy weight, current weight, and weeks of pregnancy in the baseline survey and current weight and weeks of pregnancy at the follow-up survey. Women in the txt4two group also answered questions about their program experience in the follow-up survey. 

Dietary quality was assessed with the fat and fiber behaviour questionnaire (FFBQ) [41]. The survey contains 20 items, and target fat- (13 items) and fiber- (7 items) related behaviors and are scored to create a fat index, fiber index, and total index (all fat and fiber items). To avoid making estimates from too few items, fat and fiber indices were only considered valid if at least 80% of items had non-missing responses; the total index was only valid if both indices were valid. Fruit and vegetable intake was be assessed against Australian Guide to Healthy Eating (AGHE) recommendations for pregnancy (two and five serves daily, respectively) [7].

Physical activity was assessed using pregnancy physical activity questionnaire (PPAQ) [42]. The PPAQ is self-administered and asks respondents to report the time spent participating in 32 activities, including household/caregiving, occupational, sports/exercise, transportation, and inactivity. From the PPAQ, we calculated physical activity levels (minutes/week; proportion of women meeting physical activity guidelines). Minutes of physical activity per week were assessed against Australia’s physical activity and sedentary behaviour guidelines [39], which recommend at least half an hour of moderate exercise on most, if not all, days during pregnancy (equating to 150 min per week). 

Women’s engagement with the program, including preference for initial modality and uptake of, retention in, and engagement with the program. Preference for initial goal-setting modality (dietetic F2F, dietetic led workshop/HSP, dietetic telehealth); recruitment (women engaged in program/women offered enrolment) was recorded by the program manager via survey and project database. 

### 2.7. Analysis

Quantitative data were analyzed with IBM SPSS Statistics v28 (Armonk, NY, USA: IBM Corp). Means and standard deviations (normal distributions), medians and interquartile ranges (IQR) (skewed distributions), or frequencies were calculated. The 2009 IOM GWG guidelines [6] were used in analyses. The changes in behaviors were examined over time (baseline to follow-up) between groups (intervention to control). Differences from baseline to follow-up were calculated for continuous variables. Categorical variables were constructed, reflecting proportion of women who met behavioral or GWG guidelines at each time point. Differences over time in whether women continued to meet, or not meet, guidelines were calculated. Variables for continuous measures were checked for normality assumptions. Differences were examined with independent group *t*-tests or independent group χ² tests, including Fisher’s exact tests for outcomes with cells <5, and Mann–Whitney U tests (for baseline non-parametric variables). 

## 3. Results

### 3.1. Implementation/Adoption

The program implementation commenced in February 2018 and took two years, as it was impacted by the complexities of navigating the hospital systems and by the COVID-19 pandemic. 

The adaption and adoption process required many iterations of recruitment processes and program delivery due to extensive consultation and navigation of interdependent, but unaligned, hospital systems. This included an extensive ‘review and approval’ process. This involved the expected reviews by Ethics and Governance committees, as well as information and communications technology governance (for agreement regarding website hosting, text message delivery and recruitment), management and clinicians of obstetrics, midwifery, dietetics, physiotherapy, and pregnant women (regarding content approval and updates required and program delivery), mater education (for program content), hospital marketing (for brand alignment, production of locally branded content, promotion of the research, and the txt4two program, recruitment), and research marketing (for brand alignment, branded content, promotion of the research and the txt4two program, recruitment), legal services (due to the research collaboration between interstate collaborators), and administration service (for booking and for recruitment processes). Often, negotiations with one department resulted in additional approval from other departments (management, marketing), with multiple ethics amendments required.

The project team identified enablers for the program adoption. The health service (management, clinicians, and women) embraced digital disruption, and this was viewed as an enabler because digital education resources for women and families had already been developed. The project being led by an embedded research dietitian, who worked across the clinical domain, was seen to facilitate the evidence-based adoption and allow for early, wide (and persistent) stakeholder engagement with an understanding of hospital systems. The robust implementation and co-creation plan provided a road map for implementation that was seen as a facilitator, along with the decision to be flexible and pragmatic, allowing for adaptability, when required, to fit with stakeholders and infrastructure. 

Barriers to adoption were also identified. Significant negotiations that elongated timeframes included legal agreements and intellectual property ownership and transaction across institutions and between individuals. Tailoring the content for the requirements of individual health professionals and marketing and ethics required more negotiation than expected. Navigating technology infrastructure was more complicated than anticipated. For example, gaining approval to use the system to go live took nine months, due to information technology delays and the integration being deemed a low priority. Through discussions, it was initially believed that this online process aligned with the hospital’s online intake form; however, this was later identified as being a separate process, unable to incorporate study registration, due to its own implementation trial. 

### 3.2. Reach 

Recruitment for the prospective control group ran for nine months (29 July 2019–27 April 2020) and for six months for txt4two (8 April 2020–19 October 2020), rather than the planned three months for each. Additional recruitment strategies were added during the study, in response to slow enrolment rates. Women were initially recruited via posts on the hospital’s social media accounts (Facebook, Instagram), flyers within the antenatal clinic, and via the hospital’s online registration form. An invitation text message to all eligible women registering with the hospital was introduced in June 2020. 

The number of women who registered their interest and the eligible proportions who completed each survey (and the program) are shown in Figure 1. Of the women registering their interest for the prospective cohort (n = 121), 82.9% were eligible (n = 97). For the women registering for the txt4two cohort (n = 117), only 28.9% (n = 35) were eligible, with the major ineligibilities being privately-funded hospital care (n = 50) and high gestational age (n = 18). Of the 97 eligible women enrolled in the prospective cohort, 65.9% completed survey 1 (baseline) and 45.4% completed survey 2 (follow-up) (Figure 1). Thirty-five women were eligible for txt4two and completed Survey 1; of these 42.9% (n = 15), completed survey 2 (Figure 1). These small complete data sets resulted in the study being underpowered and limited in statistical analysis.

During the study period, of the 7172 public-hospital women receiving antenatal care at the study hospital, 2100 women had a ppBMI > 25 kg/m^2^ (16.9%). The characteristics of the women who participated in each of the cohorts are shown in Table 1. Women were, on average, between 31–33 years of age, with just under two-thirds born in Australia. A higher proportion of women in the prospective control group, compared with the txt4two group, were educated at university, experiencing their first pregnancy, and identified with being aboriginal and/or a Torres Strait Islander. Women in the prospective control group had a lower pre-pregnancy weight, compared with the txt4two group, influenced by there being a greater proportion of them in the overweight category (59.4%), compared with the intervention group (45.7%) (Table 1). When compared with the general hospital population of 7172 women, of whom 28.2% (n = 1005) were in the ppBMI overweight category and 30.8% (n = 1095) were in the obese ppBMI category, they were of similar age (31.2 (5.2) years), and 3.4% and 0.4% identified as aboriginal or aboriginal and Torres Strait Islander, respectively.

The initial plan was to recruit three groups of 50 women, in order to test the different modalities of the txt4two introductory session, but this was not possible, due to the COVID-19 global pandemic. The majority of intervention group participants received their preferred modality for the introductory session, choosing telehealth delivery. However, the pivot to all care being delivered via telehealth, due to COVID-19, precluded any F2F delivery. (The majority of women in the txt4two group nominated telehealth as their preferred modality for the introductory session (73.3%; n = 22), with 16.7% (n = 5) of women nominating F2F delivery as their first preference, and 10.07% (n = 3) of women selected the Group F2F as their third preference). 

### 3.3. Effectiveness

#### 3.3.1. Engagement and Acceptability

Program engagement experiences are outlined in Appendix A. The majority of women (85.7%, n = 14) who participated in the txt4two program found it extremely (35.7%) or moderately (50.0%) useful and would definitely (64.3%) or maybe (28.6%) recommend it to a friend. The elements that were most liked about the program included the goal setting process, reminders, and encouragement, being held accountable, and being made aware of the GWG ranges. Fewer responses were noted for the least liked elements, but included a desire for phone check-ins during the program and an ability to refine goals if circumstances changed. ‘Other’ recommendations reiterated some of these points (a check-in halfway through the program and more flexibility within the program for adapting as pregnancy journey changes), as well as an online weight tracker, more clarity when they needed to respond to SMS, and more tools and resources. The number of SMS received was reported as ‘just right’ by many (71.4%), with 85.7% reporting to have read every SMS. The majority (92.9%) reported that the ‘timing’ of the SMS was acceptable. 

#### 3.3.2. Gestational Weight Gain 

As shown in Table 2, the overall GWGs in the prospective control cohort from pre-pregnancy to Survey 2 (~36 weeks) were 8.1 kg (±6.3) (n = 45) and 10.9 kg (±2.7) (n = 15) in the txt4two cohort (between-group comparison *p* = 0.02). The proportions of women meeting the IOM GWG guidelines were 11.4% (n = 4) in the control cohort, 35.3% (n = 6) in the txt4two cohort, and 4.2% (n = 1) (PC) and 33.3% (n = 4) (txt4two) for the overweight and obese ppBMI categories, respectively. The proportion of women gaining under the IOM GWG guidelines was 20% (n = 7) in the control cohort, 11.8% (n = 2) in the txt4two cohort and 33.3% (n = 8) (PC), and 16.7% (n = 2) (txt4two) for the overweight and obese ppBMI categories, respectively. The proportion of women gaining above the IOM GWG guidelines were 25.7% (n = 9) (PC), 11.8% (n = 2) (txt4two), and 20.8% (n = 5) (PC) and 8.3% (n = 1) (txt4two) for overweight and obese ppBMI categories, respectively.

#### 3.3.3. Fruit, Vegetables, and Diet Quality

A clinically meaningful increase in the servings of fruit per day was observed in the txt4two cohort, compared with the PC cohort, 0.4 (±0.9) versus 0 (±0.9); however, this did not reach statistical significance (*p* = 0.1). A significant increase in the servings of vegetables per day was observed in the txt4two cohort, compared with the PC cohort, 0.9 (±1.2) versus 0.1 (±0.7) (*p* = 0.03.) The proportion of women meeting the vegetable guidelines between groups over time approached significance (*p* = 0.06; Table 2). An improvement in the total diet quality index scores and fibre scores also improved significantly from Survey 1 to Survey 2 between groups (Table 2). 

#### 3.3.4. Physical Activity

Non-significant increases in total physical activity were observed between groups over time (+45.8 min); the sample was not powered to detect statistical differences. Non-meaningful changes were observed in minutes of sedentary time and percentage of women meeting physical activity guidelines. 

## 4. Discussion

This paper outlines the outcomes and key learnings from a pragmatic implementation of an adapted digital health program into the MMH antenatal service. The outcomes suggest that the program could feasibly be implemented with the considerable time and effort required to adapt and embed the program within the interdependent, but unaligned, hospital systems. While the recruited sample was underpowered, promisingly, clinically meaningful or statistically significant changes were observed in dietary behaviors (fruit and vegetables) and diet quality indices and overall physical activity. Greater proportions of women in the txt4two group gained within the IOM ranges for their ppBMIs. The numerous process negotiations and reflections on enablers provide guidance for future work, adapting and embedding evidence-based programs across different clinical settings. 

This research adapted an evidence-based intervention for a new implementation context. The use of adapted, previously tested interventions reduces the need to develop a new program and saves time and money, while expanding the availability of evidence-based treatments across clinical settings and demographically different populations [36]. Surface adaptations, such as those used in this study, preserve core intervention components and harmonize intervention materials to characteristics of the population and system context and have been shown to improve the outward appeal, acceptance, and face validity [36], in comparison with unadapted interventions. Despite the growth in implementation research, limited scientific attention has focused on understanding and improving the embedding and planning for sustainability of health interventions. Models of sustainability have been evolving to reflect the challenges in the fit between intervention and context. Sustainability has evolved from being considered as the endgame of a translational research process to a suggested ‘adaptation phase’ that integrates and institutionalizes interventions within local organizational and cultural contexts [43]. 

Hospitals are a series of complex administrative and clinical systems. Any intervention needs to operate through active contextual exchanges and influence, and be influenced by, system mechanisms [44]. This study outlined the complex number of negotiations (women, clinical departments and clinicians, ethics, governance, information and communication technology, education, hospital and research marketing, and legal) and reported enablers (embedded research clinician, organizational digital strategy, robust implementation plan, and a flexible, pragmatic project management approach). The key enabler identified was that the project lead researcher was a clinician embedded in, and familiar with, the hospital systems. This is consistent with emerging implementation work showing advantages of embedded clinician researchers who sit at the intersection of implementers’ appreciation of change and capacity to influence context, on the one hand, and researchers’ knowledge and expertise in empirical methods on the other [45]. The other facilitators and barriers are consistent with other implementation work [46,47,48] and highlight the complexities of embedding evidence-based work in clinical settings and systems. While these negotiations are critical to embedding programs successfully, it comes at the cost of time and money to be spent on the intervention. Further, the timelines required to achieve implementation are often at odds with the short timelines of grant funding timelines, making it difficult to demonstrate prompt success and outcomes. 

Despite the numerous barriers faced in integrating and adapting the txt4two program, as well as the lower recruitment and retention rate, it remained effective at changing the dietary behaviors and physical activity levels. The original txt4two program resulted in a lower reduction in physical activity (with over 80% of goals set relating to physical activity) [30]. It is likely that the greater focus on improving dietary quality in the goal setting worksheets and strategies, particularly fruit and vegetable intake, and offering to alternate between dietary and activity goals each six weeks impacted this outcome. Despite the lack of a significant GWG finding, the improvement of diet quality alone is important for its epigenetic effects [2,3,4,5] and potential for the reduction in chronic disease risk [27].

Future health services redesign projects that attempt to take findings from RCTs, especially related to digital health solutions, into the real world that have a number of considerations for a more streamlined approach. As new programs are embedded into services in response to health consumer needs, the enduring question of communication, recruitment, and continued engagement within education requires further refinement and research. This study showed that it took time to gain traction with recruitment and required additional, often novel (to research), avenues that aligned with usual hospital communication processes. An awareness of these avenues (by researchers and clinician–researchers), as well as a willingness to integrate programs (into hospital communication and data systems by those who manage them), is required to deeply and routinely embed interventions into business as usual for greater and more rapid impact, evaluation, and monitoring.

### Strengths and Limitations

The adaptation and testing of an effective and acceptable theoretically underpinned program that had robust formative concept testing and piloting phases was a strength of this study [30]. Furthermore, the extensive stakeholder engagement and adaptation process further enhanced the program. The limitations of the study stem from its design, using consecutively collected data. However, the use of a pragmatic implementation, while preventing randomization, provided richer insights into strategies to advance the implementation, transportability, and impact of health services research. Further limitations come from the low numbers of women who engaged in the program. Despite the study lacking sufficient power, significant improvements were still seen in dietary quality indices and vegetable intake. Whilst outcome variables relied on self-reported measures of weight, dietary intake, and physical activity, these were all valid and reliable measures and have been used in previous research [41,42]. Environmental factors, such as the COVID pandemic, seasonality, and staff changes in clinics, and external inputs, such as social media, may have affected the outcomes of the study, due to the consecutive cohort design factors) [30]. 

## 5. Conclusions

Adapting and implementing a previously tested digital intervention into a tertiary maternity hospital was feasible and elicited positive dietary quality and physical activity engagement changes. The barriers to this process included fragmented and unagile hospital and research governance and systems, while the enablers included a culture accepting of digital disruption and an embedded clinician–researcher able to navigate and overcome these complexities, while achieving diverse stakeholder engagement. As we move beyond pilots and embed research programs within health services or the community, we need to understand the facilitators and barriers to implementation and prepare implementation road maps that move beyond theory. In doing so, we may strengthen the evidence for the refinement of implementation plans. Further dialogue is also required to understand the implementation transformation of programs and the impact on program fidelity and quality and subsequent health outcomes. 

## Figures and Tables

**Figure 1 nutrients-15-00588-f001:**
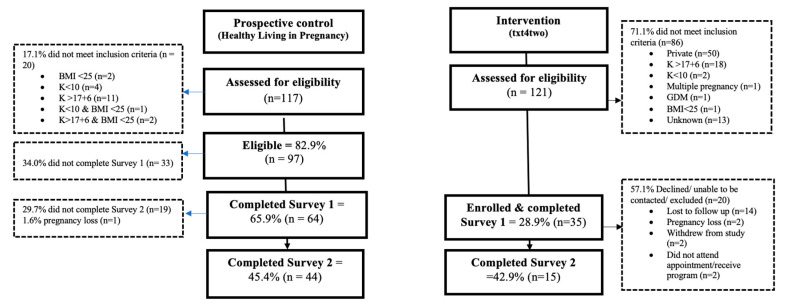
Recruitment flowchart for the prospective control and txt4two (intervention) cohorts.

**Table 1 nutrients-15-00588-t001:** Characteristics of the prospective control and txt4two (intervention) cohorts ^a^.

Characteristics	Prospective Control Cohort	txt4two Cohort
Sample size	64	35
Age (years), mean (SD)	33.2 (4.7)	31.8 (4.5)
Gestational age at study entry (weeks), mean (SD)	13.5 (3.5)	13.3 (3.1)
Gestational age at survey 2 (weeks), mean (SD)	36.5 (1.0)	36.5 (0.8)
Country of birth (Australia), n (%)	42 (65.6)	20 (57.1)
Education level, n (%)	5 (7.9)	5 (14.3)
Up to year 12	12 (18.8)	9 (25.7)
Trade/certificate/diploma university	45 (70.4)	20 (59.2)
Total yearly household income after tax, n (%)		
AUD ≤ $50,000	6 (7.2)	1 (2.0)
AUD > $50,000	63 (75.1)	39 (76.5)
Aboriginal or Torres Strait Islander status, n (%)	2 (3.1)	0 (0)
Aboriginal, but not Torres Strait Islander	0 (0)	0 (0)
Aboriginal and Torres Strait Islander Neither	60 (93.8)	34 (97.1)
Nulliparous, n (%)	36 (56.3)	15 (42.9)
Pre-pregnancy weight, kg	82.1 (13.4)	88.9 (17.8)
Weight, kg	83.4 (13.2)	90.3 (16.8)
Height, m	1.65 (6.3)	1.66 (6.5)
Pre-pregnancy BMI (kg/m^2^), mean (SD)	29.9 (4.4)	32.1 (5.5)
Pre-pregnancy BMI category, n (%)		
Overweight	38 (59.4)	16 (45.7)
Obesity	25 (39.1)	18 (51.4)

^a^ Number of participants varies due to missing data.

**Table 2 nutrients-15-00588-t002:** Behavioral and weight outcomes at Survey 1 and Survey 2 for the Prospective Control (PC) and txt4two cohorts, changes within-groups over time and differences between groups at Survey 1 and Survey 2.

Outcomes	Between-Group Difference at Survey 1 (txt4two—PC)	PC Survey 1	PC Survey 2	PC within-Group Difference over Time	txt4two Survey 1	txt4two Survey 2	txt4two within-Group Difference over Time	Between-Group Difference at Survey 2 (txt4two—PC)
Serves of fruit per day, M(SD)	*p* = 0.2	1.7 (0.8) (n = 64)	1.7 (1.0) (n = 45)	0 (0.9)	1.5 (0.9) (n = 35)	2.0 (1.0) (n = 14)	+0.4 (0.9)	*p* = 0.1
Women meeting pregnancy fruit guidelines, n (%)	*p* = 0.4	39 (60.9)	28 (43.8)	−17.1%	18 (51.4)	11 (31.4)	−20%	*p* = 0.4
Serves of vegetable per day, M(SD)	*p* = 0.8	1.5 (0.8) (n = 64)	1.7 (0.6) (n = 45)	+0.1 (0.7)	1.5 (0.7) (n = 35)	2.3 (1.2) (n = 14)	+0.9 (1.2)	*p* = 0.03
Women meeting pregnancy vegetable guidelines, n (%)	*p* = 0.3	3.1 (2)	0 (0)	−3.1%	0 (0)	2 (5.7)	+5.7%	*p* = 0.06
Diet quality— Total ‘score’, FFB (/20)	*p* = 0.07	5.9 (0.9) (n = 64)	6.1 (1.0) (n = 45)	+0.2 (0.6)	5.9 (0.6) (n = 35)	6.4 (1.0) (n = 14)	+0.7 (1.1)	*p* = 0.012
Diet quality— Fiber ‘score’ from FFB(/7)	*p* = 0.045	2.8 (0.5) (n = 64)	2.9 (0.7) (n = 45)	+0.1 (0.5)	2.5 (0.6) (n = 35)	3.2 (0.7) (n = 14)	+0.6 (0.8)	*p* = 0.008
Diet quality—Fat ‘score’ from FFB(/13)	*p* = 0.4	3.1 (0.4) (n = 64)	3.2 (0.6) (n = 45)	+0.1 (0.3)	3.0 (0.4) (n = 35)	3.1 (0.4) (n = 14)	+0.2 (0.3)	*p* = 0.2
Overall gestational weight gain, kg	*p* = 0.03	82.1 (13.4) (n = 64)	92.7 (11.8) (n = 45)	+8.1 (6.3)	90.3 (16.8) (n = 35)	100.3 (15.5) (n = 14)	+10.9 (2.7)	*p* = 0.02
Weekly minutes of TOTAL physical activity, median (IQR)	*p* = 0.5	251.2 (1040.7) (n = 62)	230.7 (936.6) (n = 44)	−8.0 (154.4)	254.7 (420.0) (n = 34)	282.5 (419.3) (n = 14)	+37.8 (76.2)	*p* = 0.3
Weekly minutes of sedentary time, M (SD)	*p* = 0.4	60.2 (14.3) (n = 62)	62.7 (15.4) (n = 44)	+2.6 (10.4)	63.3 (19.9) (n = 62)	72.4 (23.6) (n = 44)	+7.7 (22.8)	*p* = 0.4
Percentage of women meeting pregnancy physical activity guidelines, n(%)	*p* = 0.4	50 (82.0)	38 (86.4)	+4.4%	30 (88.2)	13 (92.9)	+4.7%	*p* = 0.5

## Data Availability

The data presented in this study are available on request from the corresponding author. The data are not publicly available, due to privacy issues.

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
