# Peer review of "Evaluation of the Pragmatic Implementation of a Digital Health Intervention Promoting Healthy Nutrition, Physical Activity, and Gestational Weight Gain for Women Entering Pregnancy at a High Body Mass Index"

_nutrients, 2023, doi:10.3390/nu15030588_

Round 1

Reviewer 1 Report

The manuscript is interesting, however, it suffers from two big concerns from my point of view:

1. The authors did not addressed the study design clearly enough. For example, how long did the baseline (survey 1) and follow-up (survey 2) survey being conducted? Did the two surveys share similar or different modalities of intervention? I suggest the authors provide a detailed Table clearly listed every items of the multi-modailty intervention for both control and txt4two groups, and also cleary described how every step of intervention was conducted.

2. Beyond the outcome evaluation, the manuscript should also provide a detailed process evaluation for evaluating how the intervention were conducted by the researchers and how the participants acted to the intervention.

Minor concerns:

1. Please have more disccusion on the large number of loss of follow-up.

Reviewer 2 Report

Your research is a very interesting topic as Evaluation of the pragmatic implementation of a digital health intervention promoting healthy nutrition, physical activity and gestational weight gain for women entering pregnancy at a high body mass index’.

This seems to be a good paper to inform us that the recent rapid changes in the digital healthcare environment can have a great impact on the perinatal period as well. To review this paper was very interesting work to me.

The manuscript is set up correctly. the experimentation is adequate and the results are clearly presented, the statistics correct. 

Method>

1.      A smaple size is somewhat small, but I hope that this ‘txt4two’ program will be operate and additional reports will be continue. 

2.      This paper has several strength. Questionanaires based on own people have been used such as ‘Australian Dietary Giudelines’ and ‘Fat and fibre behaviour questionanaire’. It is thought that this study will provide more accurate data as the participants' weight and dietary control are efficiently performed.

3. It was good at reviewing through provided URL, https://www.matermothers.org.au/antenatal-and-pregnancy-services/nutrition-and-ie- 197tetics/txt4twoprogram. Becasce I look the general enroll method and working program of ‘txt4two’, I have seem to understand it more.

Results>

Thank you that I have a chance with reviewing your amazing study.

I have some questions.

1. As shown your reports,  participants with pre-regnancy BMI >25kg/m2(10-17eeks) enrolled. Does they have the differences on GWG, PA, vegetable index, and fiber-diet quality index between groups according to pre-pregnancy BMI level(≤ 25, > 25, and >30)? 

If you have the analysic data about that, I hope you descript in the text.

2. I wonder you have checked the perinal outcomes of babies. Your program must be a very sensational item and I hope it helps to improve the babies’ good prognosis.

Thank you.

I look forward to futher study.
